# Production of β-Glucans from *Rhizopus oryzae* M10A1 by Optimizing Culture Conditions Using Liquid Potato Starch Waste

**DOI:** 10.3390/polym17091283

**Published:** 2025-05-07

**Authors:** Miguel Anchundia, Gualberto León-Revelo, Stalin Santacruz, Freddy Torres

**Affiliations:** 1School of Food Engineering, Carchi State Polytechnic University, Tulcán 040101, Ecuador; gualberto.leon@upec.edu.ec (G.L.-R.); freddy.torres@upec.edu.ec (F.T.); 2School of Agroindustrial Engineering, Universidad Laica Eloy Alfaro de Manabí, Manta 130222, Ecuador; stalin.santacruz@uleam.edu.ec

**Keywords:** β-glucans, *Rhizopus oryzae*, optimization, liquid potato starch waste, polysaccharides

## Abstract

β-glucans from filamentous fungi are important for human health. There is limited research on polysaccharides from filamentous fungi, and no reports have been published regarding the optimization of culture media to produce β-glucans from *Rhizopus oryzae* using liquid waste from potato starch processing. In this regard, the fermentation conditions to produce β-glucans from *Rhizopus oryzae* M10A1 were optimized using the one variable at a time (OVAT) and response surface methodology (RSM). The β-glucans were chemically characterized by determining moisture, nitrogen, protein, fat, ash, and total carbohydrates. The color, molecular weight, β-glucan content, monosaccharide composition, and structural and conformational characteristics were assessed by colorimetry, gel permeation chromatography, high-performance liquid chromatography, and Fourier transform infrared spectroscopy, respectively. The microbial indicators, mesophilic aerobes, molds, yeasts, and *Escherichia coli* were quantified following ISO standard protocols. Optimization indicated that supplementation with 0.8% (*w*/*v*) glucose and ammonium sulfate enhanced heteroglycan production (3254.56 mg/100 g of biomass). The β-glucans exhibited high purity, a light brown color, a molecular weight of 450 kDa, and a composition predominantly consisting of glucose and galactose. These findings suggest that β-glucans from *Rhizopus oryzae* M10A1 could be used for food and health applications.

## 1. Introduction

Currently, a significant portion of the world’s population suffers from gastrointestinal tract diseases, such as diarrhea, stomach pain, and colonic inflammation, which are mainly caused by dysbiosis or micro-obesity, a condition known as dysbiosis or micro-obesity [1,2].

Numerous health consequences linked to diabetes and dyslipidemia, which can result in cardiovascular disease and other issues, have been directly linked to dysbiosis. Sedentary lifestyles and poor eating habits are other variables linked to chronic diseases, which can cause a lot of deaths and high costs for healthcare systems. By 2022, these illnesses ranked among the top four global causes of death for the reasons outlined above [3].

In the context of helping to strengthen the body and restore balance in non-communicable diseases such as diabetes and dyslipidemia, special attention must be directed to research on new ingredients and food development [4,5]. In this regard, β-glucans have gained considerable interest because of their potential applications in the development of functional foods, which have been shown to promote health and help prevent various pathological conditions [6].

β-Glucans are polysaccharides composed of D-glucose monomers bound by β-glucoside linkages. The structures of fungal glucans differ in terms of their molecular structure, ranging from linear α-(1-3) and α-(1-4) linkages for α-glucans to linkages with β-(1-6) and β-(1-3) branches for β-glucans [2,7], resulting in beneficial effects on human health, including immunomodulatory and anticancer effects, maintaining glucose metabolism homeostasis, and lowering triglycerides, cholesterol, among others [7,8,9,10].

These polymers can be classified into two categories: homoglycans, composed of a single type of monosaccharide, and heteroglycans, which consist of various monosaccharides, including galactose, xylose, mannose, fucose, fructose, and arabinose [11].

Different members of the fungal kingdom have glucan contents ranging from 0.21% to 65% (*w*/*w*), depending on the species from which they are derived [12]. The most studied have been *Agaricus brasiliensis*, *Agaricus bisporus*, *Coprinus comatus*, *Laetiporus sulphureus*, *Pleurotus ostreatus*, *Ramaria botrytis*, and *Termitomyces eurrhizus*, followed by the yeasts *Saccharomyces cerevisiae* and *Saccharomyces eurhizus* and to a lesser extent filamentous molds such as *Aspergillus niger*, *Aspergillus fumigatus*, *Aspergillus terreus*, *Aspergillus nidulans*, *Aspergillus wentii*, and *Fusarium solani* DO7 [13,14].

The extant research on optimizing culture media at the laboratory or pilot scales for producing β-glucans from *Rhizopus oryzae* for use in functional foods is limited. This scarcity of research is likely due to the fact that these genera can cause diseases in humans by contaminating food and animal feed and producing mycotoxins [13,15].

Edible filamentous fungi with nutritional potential, such as *Aspergillus oryzae*, *Fusarium venenatum*, *Monascus purpureus*, *Neurospora intermedia*, and *Rhizopus oryzae*, have been used to produce proteins. Notably, *Rhizopus oryzae*, a non-pathogenic fungus, is regarded as a probiotic and is employed in the production of fermented foods. It has even been designated as generally recognized as safe (GRAS) by the Food and Drug Administration (FDA) [15,16].

In addition, extracting starch from different plants uses a lot of water, which is usually released into the environment. This can cause problems in soils and water bodies because it releases a lot of carbohydrates, protein residues, and other nutrients. Managing starch residues effectively is essential because they can support sustainable development goals 6, 12, 13, 14, and 15 of the UNESCO 2030 agenda [17,18]. The nutrients these residues provide can be used in culture media to produce filamentous fungal biomass, facilitating the extraction of β-glucans.

Liquid potato starch waste has been used for biomass production, lipid biosynthesis, nutrient biodegradation, and carotenoid production by *Candida inconspicua*, *Debaryomyces hansenii*, *Kluyveromyces marxianus*, *Kazachstania unispora*, *Zygotorulaspora florentina*, *Aspergillus oryzae* 448, *Aspergillus niger* 334, *Rhizopus oligosporus* 2710, and *Rhodotorula glutinis* [19,20,21,22,23].

Therefore, the goal of the investigation was to use liquid potato starch waste to optimize the culture conditions for *Rhizopus oryzae* M10A1 submerged fermentation. This was achieved by employing the methodology of one variable at a time (OVAT) and response surface methodology (RSM). The culture medium was characterized by chemical analysis, and microbiological, structural, physical, and chemical tests were performed on the β-glucans that were obtained.

The results showed that the polysaccharide content increased threefold when the culture medium parameters were optimized with 0.8% (*w*/*v*) glucose and ammonium sulfate. The β-glucans obtained were characterized by high purity, light brown heteroglycans, and adequate microbiological quality, suggesting their use potential in the development of functional foods and health applications.

## 2. Materials and Methods

### 2.1. Materials

The *Rhizopus oryzae* M10A1 strain was previously isolated and identified at the Laboratory of Biotechnology and Microbial Control of the Universidad Politécnica Estatal del Carchi (UPEC), and the liquid potato starch waste was supplied by UPEC’s Starch Processing Factory, homogenized, and stored at −20 °C until it was required. In this study, analytical-grade reagents were employed.

### 2.2. Inoculum Preparation

*Rhizopus oryzae* M10A1 spores were used, which were grown on potato dextrose agar (PDA) plates for 5–7 days at 27 °C. Six milliliters of sterile water were added to the plates to create the pre-inoculum, and the spores were then manually shaken with an inoculating loop. In submerged fermentation, the medium was inoculated with a spore suspension containing 5 × 10^6^ spores/mL [18,24].

### 2.3. Culture Conditions in Submerged Fermentation

Volumetric flasks with a capacity of 250 mL, covered with cotton, were utilized. These flasks contained 50 mL of liquid potato starch waste with and without supplementation. The mixture was sterilized at 121 °C for 20 min. Once the medium had been cooled, it was inoculated with 1 mL spore suspension containing 5 × 10^6^ spores/mL [16].

To optimize the process, the one variable at a time (OVAT) method was employed, with the variables of cultivation time, pH, temperature, nitrogen source, and carbon source being considered. The conditions for each variable are described in Section 2.13, entitled “Experimental design and statistical analysis”.

### 2.4. Determination of the Chemical Composition of the Culture Medium

The amount of soluble solids, suspended solids, and total solids were measured. A total volume of 10 milliliters of the homogenized samples was dried in porcelain dishes at 105 °C until a constant weight was achieved in order to determine the total solids content in the culture liquid media. By centrifuging 50 mL of culture medium sample homogenized at 3000× *g* for 10 min, the content of the suspended particles was determined. The sample was then dried at 105 °C in porcelain plates until a consistent weight was achieved. The soluble solids content was calculated by the difference between the total solids and the suspended solids. To determine the ash content, the total solids were placed in a muffle furnace at 575 °C for 5 h [18].

The nitrogen content determination was carried out using the Kjeldahl method, in which 10 mL of the culture media were collected in digestion tubes in triplicate, 15 mL of H_2_SO_4_ (concentrated sulfuric acid), and 2 Kjeldahl tablets were used. After that, the tubes with the reagents were put in the digesting unit (Tecnal Brand, São Paulo, Brazil) and left there for an hour at 420 °C. A 250 mL Erlenmeyer flask containing 4 drops of Tashiro’s indicator and 25 mL of H_3_BO_3_ (boric acid) at 4% *w*/*v* was then used to distill the nitrogen content using a distillation apparatus (Tecnal Brand, Tecnal, São Paulo, Brazil) [25].

The iodine method was used to measure the total starch content, 100 µL of each culture medium was mixed with 10 mL of 52% HCLO_4_ (perchloric acid), mixed to dissolve, and left to rest for 10 min, and distilled water was added to the volumetric flask to reach a volume of 50 mL.

Four 50 mL volumetric flasks containing 2.5 mL of 52% perchloric acid (HCLO_4_), 1.0 mL of the sample, and distilled water were prepared. A total volume of 10 mL was removed from each flask, and 0.5 mL of the iodine–iodide solution was added. The mixture was then allowed to rest for ten minutes in darkness.

Subsequently, the absorbances were measured at 600 nm with a spectrophotometer against a blank and recorded. A blank was prepared by incorporating all reagents without dissolving the sample. An additional blank was prepared from the samples without the addition of the iodine solution, and the resulting readings were corrected to ensure the presence of excess iodine–iodide solution.

A standard curve was prepared with 0.1 g of starch, which was mixed with 10 mL of HClO_4_ (perchloric acid) at 52%, and the process was continued until the spectrophotometer reading at 600 nm was obtained [26].

The total sugars were evaluated by the sulfuric phenol method, and the reducing sugars were determined by the dinitrosalicylic acid (DNS) method. For total sugar determination, 2 mL of culture medium sample was mixed with 2 mL of 5% phenol and placed in a rack immersed in a cold water bath. Five milliliters of H_2_SO_4_ (sulfuric acid) were added, and the tubes were allowed to stand for 15 min. The tubes were then measured with a spectrophotometer at a wavelength of 490 nm. The same treatment was carried out for the blank with distilled water.

The reducing sugars were determined by mixing 0.5 mL of each media with 0.5 mL of the DNS reagent, boiling for five minutes in a water bath, and stopping the reaction with a water and ice bath. The samples were reconstructed with 5 mL of distilled water, shaken, and left to rest for 15 min, and their absorbance was determined at 540 nm. The same treatment was carried out for the blank with distilled water.

A standard curve was generated using glucose concentrations ranging from 10 to 90 milligrams per liter, comprising a minimum of five data points, and the concentrations of total and reducing sugars were calculated with the standard curve [27].

### 2.5. Determination of the Biomass Concentration

The biomass was washed with deionized water repeatedly until it was free of culture broth. Subsequently, it dried at 105 °C until it reached a constant weight. After cooling, the biomass was weighed. The biomass was harvested in 48-h intervals [28].

### 2.6. Extraction of β-Glucans

The dry mycelium was disintegrated in a rotor mill (Tecnal rotor brand R-TE-651/2, Brazil). The resulting material was immersed in 95% ethanol for 3 h to remove fat via a Soxhlet apparatus. The mixture was subsequently washed with hexane. The defatted residue was dried and extracted with water at a 1:40 (*w*/*v*) ratio, 121 °C, 15 PSI, 1 h, 3 times, and filtered through a 72 µm nylon sieve.

The filtrate was subsequently subjected to a centrifugal process at 2000 × g for 15 min in a tabletop centrifuge (HERMLE brand, Model Z206A, Hermle Labortechnik, Berlin, Germany). The resulting supernatant was then concentrated to one-fifth of its original volume and precipitated for 24 h at 4 °C using 80% ethanol at a 1:6 ratio (*v*/*v*). After that, the precipitates were collected after being centrifuged again for 15 min at 2000× *g* [14,29].

### 2.7. Purification of β-Glucans

The precipitates obtained were washed with acetone-ether and 99% ethanol at a ratio of 1:3 (*w*/*v*) in successive order. Subsequently, a deproteinization process was carried out with a mixture of chloroform and n-butanol (CHCl_3_-n-BuOH) at a volume ratio of 5:1 (*v*/*v*) five times. Following this, the mixture was precipitated with 99% ethanol, dialyzed with a 12–14 kDa membrane against distilled water for 72 h, and then lyophilized [14,24].

### 2.8. Performance Determination

The yield was calculated via Equation (1).(1)$$\;\mathrm{Extraction}\;\mathrm{yield}\;\left(\%\right)=\frac{P1}{P}\times 100,$$
where P1 is the dry weight of extracted β-glucans, and P is the weight of the mycelium of the filamentous fungus [29].

### 2.9. Chemical Characterization

The moisture content was assessed using method 925.10, the total nitrogen content was determined via the Kjeldahl method (method 2001.11), and the crude protein content was calculated by multiplying the total nitrogen content by 6.25. The crude fat content (method 2003.06) and ash content (923.03) were used [30]. The total carbohydrate content was determined by calculating the difference between 100 and the sum of moisture, total fat, total protein, and ash.

### 2.10. Structural Characterization

The molecular weight, proportion of β-glucans, composition of the monosaccharides, and structural and conformational characteristics were analyzed [31].

The molecular weight was determined using a gel permeation chromatography (GPC, Thermoquest TSP P100, Thermo Fisher Scientific, Waltham, MA, USA) method, employing a TSK-G3000 133 PWXL column (Merck, Darmstadt, Germany), a TSP P100 pump, and a RI 150 refractive index detector. In each run, 20 µL of injected samples at a concentration of 0.5% (*w*/*v*) were utilized, and distilled water was employed as the mobile phase. The molecular weight was determined by reference to the calibration curve generated with the Dextranas T-series standard (Mw 5, 12.40, 80, 200, and 1000 kDa).

The composition of the monosaccharides and the proportion of β-glucans present in the sample were determined by high-performance liquid chromatography (HPLC) using a photodiode array detector (PDA) 2998 and a C18 XBridge column (5 µm × 4.6 mm × 150 mm) operating at 30 °C with a flow rate of 1.0 mL/min and a mobile phase of potassium phosphate buffer and acetonitrile (0.02 mol/mL, pH 6.7, 83:17 *v*/*v*). The standards used include fructose, galactose, mannose, glucose, arabinose, and xylose. The standard for determining the molecular weight was β-(1-3)/(1-6)-glucan from *Saccharomyces cerevisiae*. The detection was performed at 245 nm.

The sample, at a concentration of 10 mg/mL, was previously hydrolyzed with 200 µL of trifluoroacetic acid (4.0 M) at 110 °C for 2 h. The mixture was subsequently evaporated to dryness under reduced pressure. The hydrolyzed residue was dissolved in 100 µL of NaOH solution (0.6 M) with continuous heating for 1 h at 70 °C after adding 100 µL of a methanol/1-pheny-3-methyl-5-pyrazolone (PMP) solution (0.5 M). The solution was cooled to room temperature, neutralized with 100 µL of HCl (0.3 M), and dissolved in 1.7 mL of distilled water and 2 mL chloroform. The organic phase was discarded after the extraction process, which was repeated three times. Subsequently, the aqueous phase was filtered through a 0.22 µm micropore filter sheet and stored at 4 °C for HPLC analysis, as previously indicated.

Structural and conformational identification was carried out via Fourier transform infrared spectroscopy (FT-IR). Total attenuated reflection IR absorption spectra (% R) were obtained between 399.675 cm^−1^ and 4000.12 cm^−1^ using an FT/IR-4700 type A spectrophotometer (Jasco, Easton, MD, USA), with a resolution of 2 cm^−1^. The specific rotation was [α] 45 D (c 1 M, sodium hydroxide) and was measured at 589 nm in a TGS detector (Jasco, Easton, MD, USA) [32].

### 2.11. Color Determination

The color was determined using a colorimeter (CR-20, Konica Minolta Sensing Inc.,Konica, Osaka, Japan). The measured parameters were L^∗^ = luminosity (0 = black, 100 = white), a^∗^ (−a^∗^ = green, + a^∗^ = red), and b^∗^ (−b^∗^ = blue, +b^∗^ = yellow) [33].

### 2.12. Microbiological Evaluation

Microbiological determination was carried out following the protocol described in the Standards ISO, including mesophilic aerobes [34], molds, yeasts [35], and *E. coli* [36].

In the case of mesophilic aerobes, 1 mL of the original sample, if liquid, or 1 mL of the initial suspension (10^−1^), in other cases, was inoculated in three empty and sterile Petri dishes. Subsequently, 1 mL of the 10^−1^ dilution (in the case of liquids) or 1 mL of the 10^−2^ dilution (in the case of solid foods) was added to another pair of empty and sterile Petri dishes. This process was repeated for the remaining dilutions, and 12 to 15 milliliters of plate count agar (PCA) was poured into each plate, which was cooled between 44 and 47 °C.

The contents of the plates were then carefully mixed. Following this, a coating layer of approximately 3 mm molten agar was added, and the mixture was cooled to 44–47 °C. The plates were incubated at 30 ± 1 °C for 72 ± 3 h and plates with 15–300 colonies were counted.

The cultivation of molds and yeasts was carried out in three empty and sterile Petri dishes, with 15–17 mL of Dichloran Glycerol Chloramphenicol Agar (DG18) added at 45–47 °C and allowed to solidify. Subsequently, 0.1 mL of the sample was transferred if it was liquid or 0.1 mL of the initial suspension in the case of other foods. Subsequently, 0.1 mL of the following dilutions were poured on three sterile Petri dishes with DG18 agar using a sterile pipette.

The liquid was spread on the surface agar with a Drigalsky loop until absorbed. The plates were incubated in an inverted position at 25 ± 1 °C for 5–7 days, then the plates containing fewer than 150 colonies were counted. To ensure the sterility of the controls, these were also prepared in sufficient numbers and incubated in an inverted position at 25 ± 1 °C for 5–7 days.

For the *Escherichia coli* count, 1 mL of the initial sample was transferred with a pipette to empty and sterile plates if it was liquid or 1 mL of the initial dilution (10^−1^) in the case of other products, in triplicate on sterile plates of 90 mm, sterile, and 15 mL of Tryptone Bile X-glucuronide (TBX) agar tempered at 44–47 °C in a thermal water bath was poured.

The inoculum was carefully mixed with the medium, and the Petri dishes were allowed to solidify on a horizontal surface. The time elapsed between the distribution of the inoculum on the plate and the pouring of the medium should not exceed 15 min. Plates were incubated at 44 °C for 18–24 h in an inverted position, with a total incubation time of no more than 24 h. Typical *E. coli* blue/blue–green colonies were counted on plates with fewer than 150 typical colonies.

### 2.13. Design of the Experiment and Statistical Analysis

The process was optimized using one variable at a time and response surface methodology with a central compound design. The conditions for cultivation, time, pH, carbon source, nitrogen, and source concentrations of carbon and nitrogen to produce higher amounts of β-glucans were selected [37,38]. The carbohydrate sources utilized included starch, glucose, molasses, and sucrose, while the nitrogen sources used included yeast hydrolysate, urea, ammonium chloride, and ammonium sulfate. The optimization was carried out by the Design Expert 13 (2021) program, trial version.

Tests were performed in triplicate, and these data were subjected to descriptive statistics. Means ± standard deviations were calculated, and one-way analysis of variance (ANOVA) was used for comparisons, with minimum significant difference post hoc statistics. The level of confidence used was 95%. In both cases, the software Statgraphics Centurion (2024), a trial version, was used.

## 3. Results and Discussion

### 3.1. Determination of Submerged Fermentation Parameters

#### 3.1.1. Determination of Time and Temperature

Figure 1 presents the incubation time and temperature required to produce β-glucans from *Rhizopus oryzae* M10A1. The highest quantity of β-glucans was produced at 30 °C after 4 days of submerged fermentation (413.16 mg/100 g of biomass), decreasing later until the lowest quantity (81.12 mg/100 g of biomass) was reached at 40 °C after 14 days of culture.

In *Fusarium solani*, the impact of temperature on the synthesis of polysaccharides was examined at temperatures ranging from 20 to 30 °C. The results indicated that 28 °C was the optimal temperature, thereby confirming that temperature plays an important role in the enzymatic induction of polysaccharide production [38].

The optimal duration for the highest production of polysaccharides has been reported among different microorganisms. For *Fusarium solani,* it is between 10 and 14 days; for *Aspergillus oryzae* and *Rhizopus oryzae,* it is 2 days [18,38,39]. These differences can be attributed to the utilization of different microorganisms and substrates.

#### 3.1.2. Determination of pH

Table 1 shows the results for obtaining β-glucans according to the effect of pH. The highest amount of β-glucans was obtained at pH 6, with a value of 86.62 mg/100 g of biomass.

The pH values reported for the biomass and β-glucan production differ; for example, pH 6 for *Rhizopus oryzae*, pH 3.85 for *Aspergillus oryzae*, *Fusarium venenatum*, *Monascus purpureus* and *Neurospora intermedia*, and pH 6.50 for *Fusarium solani*. In both cases, pH values below and above the optimum have been shown to decrease the concentration of these compounds because of the effects on the activity of the enzymatic machinery and the availability of metabolic pathways [18,38,39].

#### 3.1.3. Determination of the Carbon Source

With regard to the carbon source, it was observed that higher concentrations of β-glucans were obtained with glucose as the carbon source (244.94 mg/100 g of biomass) compared with the control (203.97 mg/100 g of biomass). The glucose concentration that yielded the maximum amount of β-glucan was determined to be 0.6%, yielding values of 128.77 mg/100 g of biomass (Figure 2). It is evident that as glucose concentrations increase, polysaccharide production undergoes a decline.

In other investigations, glucose promoted the growth of *Lentinus edodes and Rhizopus oryzae* and increased the concentration of polysaccharides of *Fusarium solani* [38,40,41,42]. Regarding the sources of carbon, carbohydrates are the main source of energy, constitute an essential component of the cell cytoskeleton, and are directly related to the growth and development of fungi and metabolite production [38].

Glucose is a more effective biological energy source than other carbon sources because it is metabolized more efficiently and is less expensive. This makes it a more viable option for use in culture media designed with residues from agro-industrial processes. Consequently, this carbon source is preferable for supplementing culture media to produce biomass and polysaccharides [43].

Excessive use of carbohydrates can have negative effects on carbohydrate metabolism, such as suppressing catabolism and reducing the synthesis of enzymes involved in sugar catabolism [40].

These findings can help explain the behavior of *Rhizopus oryzae* M10A1 with the source of carbohydrates used in the study, as well as a reduction in the amounts of β-glucans obtained when the culture medium is supplemented with glucose at concentrations greater than 0.6%

#### 3.1.4. Determination of the Nitrogen Source

The fungus *Rhizopus oryzae* M10A1 was cultivated in a variety of nitrogen sources; however, ammonium sulfate was the source that allowed for greater amounts of β-glucans (244.83 mg/100 g of biomass), and the lowest polysaccharide content was obtained with yeast extract (82.21 mg/100 g of biomass). The most effective ammonium sulfate concentration for generating higher levels of β-glucans was determined to be 1.0%, yielding results of 388.67 mg/100 g of biomass (Figure 3).

The supplementation of culture media with various nitrogen sources has been demonstrated to exert a positive effect on the production of polysaccharides in fungi. The utilization of diverse nitrogen sources, including ammonium sulfate, meat extract, peptone, yeast extract, baking powder, ammonium nitrate, glycine, ammonium chloride, and urea, has yielded favorable outcomes. In particular, yeast extract has been identified as a particularly effective nitrogen source for various fungal types, with reported concentrations ranging from 0.5% to 0.7%; this has led to the production of polysaccharide contents ranging from 0.32 to 1.87 g/L [38,40]. However, the use of ammonium sulfate as a supplement for the cultivation of *Rhizopus oryzae* resulted in increased biomass production (29.50 g/L) [44].

The results obtained in this study are consistent with those of a previous study, where ammonium sulfate was the most efficient nitrogen source for *Rhizopus oryzae*. This suggests that nitrogen is crucial for producing polysaccharides and that its effects are affected by several factors, such as microorganism type, nitrogen source, concentration, and culture type. High nitrogen concentrations can reduce other carbon sources and the accumulation of metabolites [40].

### 3.2. Optimization of Growth Conditions

The experimental arrangement for optimizing *Rhizopus oryzae* M10A1 β-glucan production based on the ascending scale approach and the central compound design model is shown in Table 2. The highest values of β-glucans were obtained at two points of the cube and two axial points (runs 2, 6, 14, and 18). Specifically, in run 2, a maximum of 3218.67 milligrams of β-glucans per 100 g of biomass was achieved.

The *p*-values of each coefficient evaluated are shown in Table 3. Since these have values less than 0.05, they indicate that the coefficients are significant for the quadratic model evaluated. Consequently, the optimization equation incorporates two linear coefficients, two squares, and one interaction.

On the basis of Table 3, the response predicting the production of β-glucans (Y_β-glucans_) is given by regression Equation (2) in coded units:Y_β-glucans_ = 1206.20 + 406.40X_1_ − 310.5X_2_ + 619.70 X_1_^2^ + 246.80 X_2_^2^ − 465.0 X_1_ × X_2_,(2)

The F-test value was 56.28 with a *p*-value of less than 0.05, indicating that the quadratic regression model is significant. The adjusted coefficient of determination (R2adj) was 92.32%, indicating a strong correspondence between the predicted values and the quadratic regression model, suggesting a high degree of correlation between the experimental and predicted values.

This finding indicates that 92.32% of the observed variation in the content of β-glucans extracted from *Rhizopus oryzae* M10A1 can be attributed to the second-order polynomial prediction equation. The value of the precision index obtained was 20.69, which is higher than 4.00. This indicates that the model can be used to navigate the space of the experimental design. The f-value with a lack of fit was 2.10, with a *p*-value of 0.143, suggesting that the loss of fit was due to the error being not significant. This outcome suggests a high degree of accuracy and consistency in the experimental values, thereby confirming the adequacy of the selected model.

Figure 4 shows the effects of glucose and ammonium sulfate concentrations on β-glucan production from *Rhizopus oryzae* M10A1 in two dimensions (2D) and three dimensions (3D). The highest recorded polysaccharide concentration (3218.67 mg/100 g of biomass) was achieved through the utilization of 0.8% glucose and 0.8% ammonium sulfate. Furthermore, the surface obtained is a quadratic and not linear plane.

As evidenced by the response surface (3D Graph) and contour plots (2D Graph), the changes in color are indicative of varying combinations of ammonium sulfate and glucose concentrations and their impact on the production of β-glucans. Specifically, the intensity of the red color, observed at a concentration of 0.8%, indicates the highest polysaccharide concentration obtained.

The validation of this model was carried out under the established conditions, producing a concentration of β-glucans of 3200 mg/100 g of biomass. A good correlation was observed between the expected values and those predicted by the model.

### 3.3. Chemical Characterization of the Culture Medium

Appendix A present the findings of the characterization of the components of the culture media utilized in the optimization of the production of β-glucans prior to and following fermentation with *Rhizopus oryzae* M10A1. The starch content, being the most representative component of the media, varied from 3107.52 to 4342.11 mg/100 mL before culture and from 400.58 to 756.04 mg/100 mL after culture.

Total and reducing sugars were present in lower proportions. Prior to and following cultivation, the total sugar concentration varied between 115.38 and 390.01 mg/100 mL and 5.85 and 90.12 mg/100 mL, respectively. Similarly, the levels of reducing sugars exhibited variability, ranging from 43.11 to 67.29 mg/100 mL before and after culture and from 4.60 to 18.13 mg/100 mL, respectively.

In regard to culture medium run No. 2, in which the highest concentration of β-glucans was obtained, it was observed that 88.27% of the starch was consumed, whereas 61.89% of total solids, 42.95% of suspended solids, 79.07% of soluble solids, 5.88% of ashes, 44.68% of nitrogen, 92.07% of total sugars, and 75.05% of reducing sugars were consumed.

The values of starch and reducing sugars are found to be lower than those reported in other investigations. In these investigations, concentrations between 12.50 and 62.7 mg/L and 0.25 and 2.29 g/L were indicated for liquid residues from the potato starch production process [39]. Furthermore, total, suspended, and soluble solids are also lower in liquid waste from the wheat starch industry [18].

In addition, the ash and nitrogen content values are lower than those reported in this investigation since 2.91% and 2.98% values were found, respectively, in the liquid waste produced by pea processing [16]. The observed discrepancies may be attributed to the varying industrial processes from which the liquid waste was obtained, as well as to variations in the nutrient content of the plant sources utilized [18,25].

### 3.4. Yield and Characterization of β-Glucans

#### 3.4.1. Performance and Chemical Characterization

Table 4 shows the results of the yield and chemical characterization of the β-glucans of *Rhizopus oryzae* M10A1. The yield of the extraction process of β-glucans extracted from the cell walls of *Rhizopus oryzae* was 3.42%.

The β-glucans of *Rhizopus oryzae* M10A1 presented a high value of total carbohydrates (82.46%), followed by the moisture content (11.21%). The lowest values corresponded to the ash, nitrogen, protein, and total fat components. These values suggest that the β-glucans obtained are mainly composed of sugars.

The yield obtained for the extraction of β-glucans is comparable with the information reported for fungi belonging to the genera *Aspergillus*, *Ganoderma*, and *Pleurotus*, where values between 1.53 and 57% were obtained [45]. In contrast, values in the range of 0.7–23.8% were reported for filamentous fungi, and a lower content (0.9%) of β-glucans was obtained from the cell walls of the mycelium of *Rhizopus oryzae* [46]. In this study, the amount of these polysaccharides was increased threefold by optimizing the culture medium on the basis of liquid potato starch residue supplemented with glucose and ammonium sulfate.

The observed variations in extraction performance can be attributed to various factors, including the species from which the polysaccharides are extracted, the chemical composition of the polysaccharides, the metabolic processes involved in their synthesis, and the presence of other compounds [29,47,48,49,50].

The contents of total sugars, nitrogen, proteins, and lipids are similar to those reported for β-glucans extracted from *Laminaria japonica* and *Fusarium solani* [14,29]. The contents of these components are indicators of purity; in this sense, the content of total sugars must be greater than 80%, and that of other components must be minimized [51]. According to this information, the β-glucans obtained were of adequate purity.

#### 3.4.2. Physical Characterization

Concerning the physical characteristics, the color of the β-glucans is light brown, as illustrated in Figure 5a. The color coordinates on the CIELAB scale are L: 33.42, a: 3.67, and b: 17.44 (Table 5). On this scale, the color tends more toward red and yellow and is more luminous.

The light brown color of β-glucans has been reported in the literature as a characteristic of *Laminaria japonica*, *Fusarium solani*, and *Avena sativa* [52]. The luminosity value (L) was lower, and the chromaticity values were similar to those reported for β-glucans from barley and *Agaricus bisporus* [33,53]. This color does not induce major changes and, therefore, does not exclude its industrial use in light or transparent foods [8].

#### 3.4.3. Structural Characterization

Table 6 shows the results of the monosaccharides present in the β-glucans extracted from *Rhizopus oryzae* M10A1. The most abundant monosaccharides were glucose (59.84%), galactose (9.01%), and fructose (8.61%), and the least abundant was arabinose (4.80%). The polysaccharide had a molecular weight of 450 kilodaltons (kDa).

Glucose has been reported as the main monosaccharide in β-glucans of filamentous fungi, such as *Aspergillus *spp., *Aspergillus terreus*, *Aspergillus fumigatus*, *Aspergillus nidulans*, *Aspergillus niger*, and *Aspergillus wentii*, with values higher than 93.6%, with a lower proportion of mannose and, as trace elements, galactose [13,24,54].

The results indicated variations in the monosaccharide profile, attributable to the heterogeneous composition of the isolated β-glucans, which consequently fall under the classification of heteroglycans. In *Colletotrichum alatae* LCS1 and *Pestalotiopsis chamaeropsis* CEL6, these types of compounds were found in β-glucans isolated from cell walls. The major monosaccharide was glucose, followed by mannose, rhamnose, arabinose, galactose, and fucose [55,56].

The composition of monosaccharides is a critical factor that influences their biological activity. It has been demonstrated that the presence of arabinose, mannose, and glucose confers antioxidant capacity to β-glucan [57]. This observation suggests that the incorporation of these compounds into food systems may enhance antioxidant activity because of their heterogeneity in monosaccharide composition and the presence of monosaccharides with established biological activity.

The molecular weights are different from those reported in other studies; values between 25 and 50 kDa, 133.60 and 139.10 kDa, and less than 850 kDa have been reported [14,54,58].

According to conventional classification, β-D-glucans with molecular weights of up to 500 kilodaltons (kDa) are designated as low molecular weight (LMW) polymers, whereas those with molecular weights of 1000 kDa or more are classified as high molecular weight (HMW) polymers. Additionally, there is recognition of a third category, which includes values between 500 and 1000 kDa [59]. Consequently, the compounds obtained can be classified as low molecular weight.

The molecular weight of β-glucans is a critical factor in determining their biological activity. In this regard, polysaccharides with high molecular weights (>1000 Da) exhibit diminished antioxidant capacities, potentially due to their reduced ability to penetrate cell membranes. In contrast, polysaccharides with low molecular weights exhibit high antioxidant capacities [60].

Consequently, the β-glucans of *Rhizopus oryzae* M10A1 may possess an antioxidant capacity that ranges from high to intermediate. In a similar way, the antitumor activity of β-glucans with molecular weights above 100 kDa has been reported [51].

From an industrial perspective, another important aspect is the relationship between molecular weight, viscosity, solubility, and gel formation. It has been documented that high molecular weight β-glucans exhibit elevated viscosities and diminished solubility. Their physiological impact is analogous to that of dietary fiber, which is advantageous from a health perspective. Moreover, these compounds lack the capacity to form gels [61]. Low molecular weight β-glucans have the ability to reduce the glycemic response, lower LDL cholesterol levels, lower blood pressure, and prevent obesity [7]. This beneficial health effect can be achieved with low-weight *Rhizopus oryzae* β-glucans and can be evaluated in future research.

Figure 5b shows the FT-IR reflectance spectrum of β-glucans extracted from *Rhizopus oryzae*. The spectrum displays bands in the 1102.6, 1076.08, 1047.16, 918.91, and 864.92 cm^−1^ regions. Additional bands are evident within the carbohydrate region at 1153.71 and 1025.94 cm^−1^.

The bands in the region of 1160, 1078, and 1.044 cm^−1^, as mentioned above, are characteristic of β-(1-3)/(1-6)-D-glucans. The bands near 890 and 920 cm^−1^ correspond to β-(1-3) and β-(1-6) glycosidic bonds, respectively. The bands at 1025.94 and 1153.71 cm^−1^ (near 1023 and 1155 cm^−1^) are characteristic of α-(1-4)-(1-6)-D-glucans [62,63].

Therefore, despite the presence of characteristic bands of β-(1-3)/(1-6)-D-glucans, corresponding to 61.55% of the analyzed sample (Table 6), molecules of α-(1-4)/(1-6)-D-glucans may be present in the analyzed extract, accounting for 20.91% of the total carbohydrates determined (82.46%).

#### 3.4.4. Microbiological Characterization

With regard to the microbiological indicators of quality and fecal contamination, the evaluated samples exhibited an *E. coli* count of less than 1.0 × 10^1^ CFU/g, a mesophilic aerobe count of 3.3 × 10^1^ CFU/g, and a mold and yeast count of 1.0 × 10^1^ CFU/g (Table 7).

The findings of mesophilic aerobes, molds, and yeast were lower, and the indicator of fecal contamination, *E. coli*, was consistent with other investigations [64,65].

In this context, the UCF/g values of mesophilic aerobes, molds, and yeasts should not exceed 103 in food products that do not require cooking; however, these polysaccharides are used as raw materials and in a process where heat treatment is applied. *E. coli* is an indicator of fecal contamination and should have a count of <1.0 × 10^1^ CFU/g since its presence in food and raw materials indicates the possible presence of enteric pathogens that can affect the health of consumers [66].

Despite being executed at the laboratory scale, this approach yielded heteroglycans as raw materials, exhibiting adequate physical, chemical, structural, and microbiological characteristics. These characteristics suggest promising applications in the development of functional foods and the utilization of liquid waste from potato starch processing. The proposed research in this study presents an alternative for the management of this liquid waste, aiming to reduce its discharge to the environment and explore a biotechnological application using filamentous fungi, such as *Rhizopus oryzae* M10A1. From a different perspective, it becomes the starting point for scale-up to pilot and industrial scales, which deserves attention in future research.

## 4. Conclusions

The liquid residue was characterized by starch as its most representative component, with concentrations ranging from 3100.09 to 4342.11 mg/100 mL, which were consumed by 88.27% of *Rhizopus oryzae* M10A1 in the submerged fermentation process. These changes allowed for a reduction in the nutrient content in the waste, thus contributing to proper management and producing less pollution in the environment.

The optimal physical parameters for the highest yield of β-glucans were a pH of 6, a temperature of 30 °C, and a culture time of 4 days; these parameters are important for the enzymatic induction process for polysaccharide biosynthesis and vary among the various microorganisms used in submerged fermentation procedures.

The carbon and nitrogen sources optimized for the highest β-glucan production were glucose and ammonium sulfate. Glucose is the preferred carbon source because of its enhanced biological effectiveness, which is attributed to its faster metabolic rate and lower cost. Ammonium sulfate has been demonstrated to influence a variety of physiological processes in yeast, including metabolite production, fungal morphology, biomass, and β-glucan production.

The liquid medium from potato starch processing residues was optimized with glucose (0.8% *w*/*v*) and ammonium sulfate (0.8% *w*/*v*) to obtain β-glucans from *Rhizopus oryzae* M10A1 at a concentration of 3254.56 mg/100 of biomass, achieving 3.42% yield. These liquid residues can be utilized for biomass cultivation and polysaccharide production.

β-glucans were extracted from *Rhizopus oryzae* M10A1, possessing a total sugar concentration of 82.46% and having ash, nitrogen, protein, and fat values below 1%. Consequently, the resulting β-glucans are considered adequate in purity for use in functional foods and other applications.

The polysaccharides of *Rhizopus oryzae* M10A1 exhibited a light brown color, with coordinates of L: 33.42, a: 3.67, and b: 17.44 according to the CIELAB scale; these values indicate that the β-glucans are suitable for incorporation in food development since they do not impart dark tones to food systems.

β-glucans with a molecular weight of 450 kDa and a heterogeneous composition of monosaccharides were obtained, with glucose and galactose being the main components at 39.84% and 16.61%, respectively. The least abundant monosaccharides were galactose, fructose, mannose, and arabinose. The heterogeneity of these monosaccharides suggests a potential for biological activities, including antioxidant, antitumor, immune system modulation, carbohydrate metabolism, and lipid metabolism, among others.

The obtained β-glucans are of high quality and safety, as indicated by microbiological indicators, which were below the detection limit for E. coli (<1.0 × 10^1^ CFU/g) and estimated at 3.3 × 10^1^ CFU/g for mesophilic aerobes. Consequently, β-glucans, serving as fundamental raw materials, are characterized by their high quality and safety, making them suitable for utilization in the development of functional foods and other applications.

## Figures and Tables

**Figure 1 polymers-17-01283-f001:**
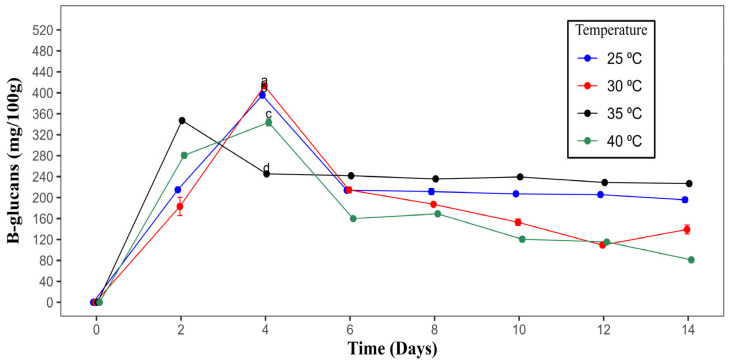
Effects of time and temperature on the production of β-glucans by *Rhizopus oryzae* M10A1. The values correspond to the average of three determinations ± the standard deviation. Different letters indicate significant differences between the production temperatures of β-glucans at a confidence level of 95%.

**Figure 2 polymers-17-01283-f002:**
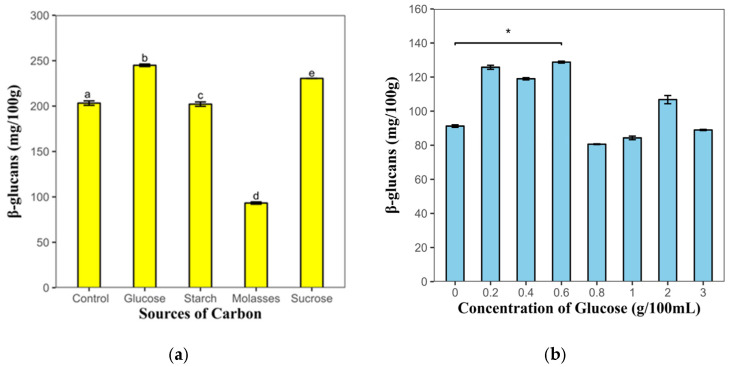
Effects of carbon source and glucose concentration on the production of β-glucans by *Rhizopus oryzae* M10A1. (**a**) β-glucans produced by the effects of carbon sources. (**b**) β-glucans produced by the effect of glucose concentration. The values represent the average of three determinations ± the standard deviation. Different letters or * indicate significant differences compared with the control (*p* < 0.05).

**Figure 3 polymers-17-01283-f003:**
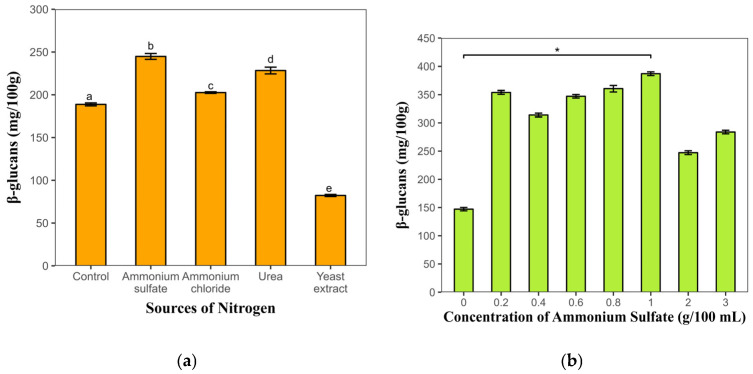
Effects of the nitrogen source and ammonium sulfate concentration on the production of β-glucans by *Rhizopus oryzae* M10A1. (**a**) β-glucans produced by the effects of nitrogen sources. (**b**) β-glucans produced by the effect of ammonium sulfate concentration. The values represent the average of three determinations ± the standard deviation. Different letters or * indicate significant differences compared with the control (*p* < 0.05).

**Figure 4 polymers-17-01283-f004:**
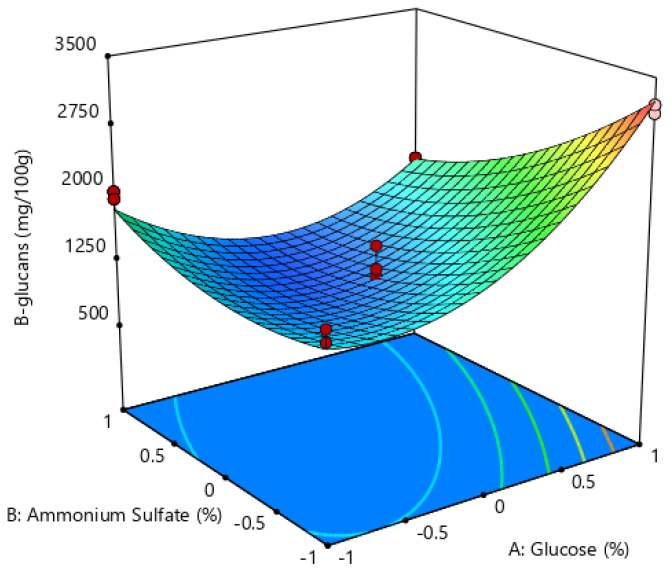
2D and 3D graphs of the effects of the concentrations of glucose and ammonium sulfate, showing the most important interactions for β-glucan production from Rhizopus oryzae M10A1.

**Figure 5 polymers-17-01283-f005:**
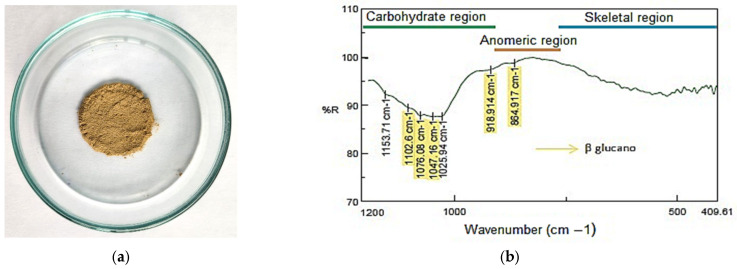
(**a**) FT-IR reflectance spectrum of the β-glucan sample from Rhizopus oryzae M10A1. (**b**) Color of the β-glucans of Rhizopus oryzae.

**Table 1 polymers-17-01283-t001:** Effect of pH on the production of β-glucans from *Rhizopus oryzae* M10A1.

Initial pH	Final pH	β-Glucans (mg/100 g) ^1^
2	3.66	0.00 ± 0.00 a
3	3.05	83.30 ± 0.74 b
4	7.36	83.74 ± 0.89 c
5	7.49	74.64 ± 1.58 d
6	7.05	86.62 ± 1.00 e
7	7.15	65.52 ± 1.44 f

The values correspond to the average of three determinations ± the standard deviation. Different letters in the same column indicate significant differences between the pH values of β-glucans at a confidence level of 95%. ^1^ Milligrams of β-glucans obtained per 100 g of mold biomass.

**Table 2 polymers-17-01283-t002:** Experimental arrangement and response of the central compound design for the optimization of β-glucan production from *Rhizopus oryzae* M10A1.

Run Order	Independent Variables		β-Glucans (mg/100 g)	
	X_1_: CG ^1^ (%)	X_2_: CSA ^2^ (%)	Measured	Predicted
1	−1 (0.4)	−1 (0.8)	1575.62	1511.67
2	1 (0.8)	−1 (0.8)	3218.67	3254.56
3	−1 (0.4)	1 (1.2)	1940.92	1820.64
4	1 (0.8)	1 (1.2)	1588.21	1703.44
5	−1.41 (0.32)	0 (1)	1787.3	1870.73
6	1.41 (0.88)	0 (1)	3143.8	3020.27
7	0 (0.6)	−1.41 (0.72)	1869.96	2138.82
8	0 (0.6)	1.41 (1.28)	1406.14	1260.49
9	0 (0.6)	0 (1)	1571.65	1206.15
10	0 (0.6)	0 (1)	1223.27	1206.15
11	0 (0.6)	0 (1)	1130	1206.15
12	0 (0.6)	0 (1)	1201.72	1206.15
13	−1 (0.4)	−1 (0.8)	1709.51	1511.67
14	1 (0.8)	−1 (0.8)	3128.55	3254.56
15	−1 (0.4)	1 (1.2)	2021.93	1820.64
16	1 (0.8)	1 (1.2)	1716.58	1703.44
17	−1.41 (0.32)	0 (1)	1574.73	1870.73
18	1.41 (0.88)	0 (1)	3116.48	3020.27
19	0 (0.6)	−1.41 (0.72)	2370.12	2138.82
20	0 (0.6)	1.41 (1.28)	992.709	1260.49
21	0 (0.6)	0 (1)	1311.49	1206.15
22	0 (0.6)	0 (1)	1192.96	1206.15
23	0 (0.6)	0 (1)	1081.01	1206.15
24	0 (0.6)	0 (1)	937.131	1206.15

The experimental design was carried out with two factors, including twelve base runs and two replications for each treatment. The tests were carried out at a temperature of 30 °C, pH of 6, stirring speed of 150 rpm, volume of culture medium of 50 mL, and Erlenmeyer volume of 250 mL. ^1^ Concentration of glucose in grams per 100 mL of culture medium. ^2^ Concentration of ammonium sulfate in grams per 100 mL of culture medium.

**Table 3 polymers-17-01283-t003:** Analysis of variance (ANOVA) of the response surface model for the optimization of the production of β-glucans from *Rhizopus oryzae* M10A1.

Source	Degrees of Freedom	Sum of Square	Contribution	Mean Squares	*F*-Value	*p*-Value
Model	5	11,032,001	93.99%	2,206,400	56.28	0.000
Linear	2	4,185,813	35.66%	2,092,907	53.39	0.000
Conc_glucose	1	2,642,877	22.52%	2,642,877	67.42	0.000
Conc_sulfate	1	1,542,936	13.15%	1,542,936	39.36	0.000
Square	2	5,116,250	43.59%	2,558,125	65.26	0.000
Conc_glucose × Conc_glucose	1	4,336,913	36.95%	4,915,134	125.38	0.000
Conc_sulfate × Conc_sulfate	1	779,337	6.64%	779,337	19.88	0.001
Interaction of 2factors	1	1,729,938	14.74%	1,729,938	44.13	0.000
Conc_glucose × Conc_sulfate	1	1,729,938	14.74%	1,729,938	44.13	0.000
Error	18	705,635	6.01%	39,202		
Lackoffit	3	208,582	1.78%	69,527	2.10	0.143
Pureerror	15	497,053	4.23%	33,137		
Total	23	11,737,636	100.00%			

R-squared (R^2^): 93.90%, adjusted R-squared (R2adj): 92.32%, precision index (adequate precision): 20.69. Conc: Concentration.

**Table 4 polymers-17-01283-t004:** Performance and chemical characterization of the β-glucans of *Rhizopus oryzae* M10A1.

Parameters	β-Glucans from *Rhizopus oryzae* M10A1
Yield	3.42 ± 0.51
Humidity	11.21 ± 0.68
Ashes	1.25 ± 0.05
Nitrogen	0.14 ± 0.01
Protein	0.89 ± 0.04
Fats	0.63 ± 0.08
Total carbohydrates	82.46 ± 0.04

The values represent the average of three determinations ± the standard deviation. The values are expressed as percentages (g/100 g).

**Table 5 polymers-17-01283-t005:** Color of the β-glucans of *Rhizopus oryzae* M10A1.

Color Parameters	β-Glucans from *Rhizopus oryzae*
L	33.42 ± 0.40
a	3.67 ± 0.06
b	17.44 ± 0.33

The values represent the average of three determinations ± the standard deviation. The units of color are dimensionless.

**Table 6 polymers-17-01283-t006:** Quantification of the monosaccharides present and the molecular weight of the β-glucans of *Rhizopus oryzae* M10A1.

Monosaccharides	Value
Xylose	8.12
Arabinous	4.80
Glucose	59.84
Mannose	7.21
Galactose	9.01
Fructose	8.61
Other	2.41
Molecular weight	450
Proportion of β-glucans	61.55

The compositional values of the monosaccharides and β-glucans are expressed as percentages, and the molecular weights are expressed in kilodaltons (kDa).

**Table 7 polymers-17-01283-t007:** Quality and safety parameters of the β-glucans of *Rhizopus oryzae* M10A1.

Parameters	β-Glucans from *Rhizopus oryzae*
Mesophilic aerobes	Estimated 3.3 × 10^1^
Molds and yeasts	Estimated 1.0 × 10^1^
*Escherichia coli*	<1.0 × 10^1^

The values represent the average of three determinations. The values are expressed in colony-forming units (CFU)/g.

## Data Availability

Data are contained within the article and Appendix A.

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
