# Peer review of "Production of β-Glucans from Rhizopus oryzae M10A1 by Optimizing Culture Conditions Using Liquid Potato Starch Waste"

_polymers, 2025, doi:10.3390/polym17091283_

Round 1

Reviewer 1 Report

Comments and Suggestions for Authors

The manuscript entitle "Production of β-glucans from Rhizopus oryzae M10A1 by opti-2 mizing culture conditions using liquid potato starch waste" is a very good study but not clearly presented. The presentation needs a lot of improvement such as 

  1. Abstract: Abstract is not comprehensive. It must have a short background, methodology adopted and results of the study. The abstract must be rewritten in a manner that all the informations are accommodated.
  2. Introduction: The Introduction is too lengthy and moves around the objectives of the study. The objectives of the study must be given in a clear manner. 
  3. Materials and methods: Well described but again need improvement of the language. In the materials and methods section, the authors have described the anlysis of media but in the results section, it is not correlated with the mycial biomass production of Rhizopus. 
  4. Results and discussions: The results and discussion section is not clearly presented and needs improvement. 
  5. Conclusions: Needs improvement in language as well as content. No mention about the temperature and pH optimization for higher b-glucan production. Effect of media composition is not correlated with the biomass production.   
Comments on the Quality of English Language

English needs to be improved substancially. 

Author Response

1. Summary

Thank you for taking the time to make revisions to the submitted manuscript, your comments have been taken into account and some clarifications have been made. The document has been updated with the changes.

2. Point-by-point response to Comments and Suggestions for Authors

Comments 1: [Abstract: Abstract is not comprehensive. It must have a short background, methodology adopted and results of the study. The abstract must be rewritten in a manner that all the informations are accommodated.

Response 1: We agree with this comment. Therefore, we have modified the abstract for greater fluidity, background and methodologies were briefly mentioned, however the methodologies were not described due to their length, these are described in the materials and methods section. Between lines 10 and 23 the following was placed: β-glucans from filamentous fungi are important for human health. There is limited research on polysaccharides from filamentous fungi, and no reports have been published regarding the optimization of culture media for the production of β-glucans from Rhizopus oryzae using liquid waste from potato starch processing. In this regard, the fermentation conditions for the production of β-glucans from Rizhopus oryzae M10A1 were optimized using the One Variable at a Time (OVAT) and Response Surface Methodology (RSM). The β-glucans were chemically characterized by determination of moisture, nitrogen, protein, fat, ash and total carbohydrates. The color, molecular weight, β-glucan content, monosaccharide composition, structural and conformational characteristics were assessed by colorimetry, Gel Permeation Chromatography, High-Performance Liquid Chromatography and Fourier Transform Infrared Spectroscopy respectively. The microbial indicators, mesophilic aerobes, molds, yeasts and Escherichia coli were quantified following ISO standard protocols. Optimization indicated that supplementation with 0.8% (w/v) glucose and ammonium sulfate enhanced heteroglycan production (3254.56 mg/100 g of biomass). The β-glucans exhibited high purity, a light brown color, a molecular weight of 450 kDa, and a composition predominantly consisting of glucose and galactose. These findings suggest that β-glucans from Rhizopus oryzae M10A1 could be used for food and health applications.

Comments 2: Introduction: The Introduction is too lengthy and moves around the objectives of the study. The objectives of the study must be given in a clear manner.

Response 2: Agree. We have modified the introduction, was summarized to emphasize the problem to be solved and the background of the research. The objectives were stated in lines 86 to 91, page 2: The results showed that the polysaccharide content increased threefold when the culture medium parameters were optimized with 0.8% (w/v) glucose and ammonium sulfate. The β-glucans obtained were characterized by high purity, light brown heteroglycans, and adequate microbiological quality, suggesting their use potential in the development of functional foods and health applications.

Comments 3: Materials and methods: Well described but again need improvement of the language. In the materials and methods section, the authors have described the anlysis of media but in the results section, it is not correlated with the mycial biomass production of Rhizopus. 

Response 3: Agree. The language has been improved for better understanding. The effect of culture media composition on production is not discussed since the variable of interest is β-glucan production. The modifications are located between lines 97 and 297.

Comments 4: Results and discussions: The results and discussion section is not clearly presented and needs improvement.

Response 4: Agree. The presentation of the results was improved, the writing was done in a clear and concise manner for a better understanding. The response variable of interest was the amount of B-glucans produced, so the information regarding biomass production was removed from the manuscript to avoid confusion.

Comments 5: Conclusions: Needs improvement in language as well as content. No mention about the temperature and pH optimization for higher b-glucan production. Effect of media composition is not correlated with the biomass production.  

Response 5: Agree. The language of the conclusions was improved in both language and content. The effect of temperature and pH on β-glucan production was included between lines 648 and 651. The content included was as follows: The optimal physical parameters for the highest yield of β-glucans were a pH of 6, a temperature of 30°C, and a culture time of 4 days, these parameters are important for the enzymatic induction process for polysaccharide biosynthesis and vary among the various microorganisms used in submerged fermentation procedures.

3. Response to Comments on the Quality of English Language

Point 1: English needs to be improved substancially. 

Response 1: The English was revised and improved throughout the text. The entire manuscript was changed to allow for better understanding of the information.

Reviewer 2 Report

Comments and Suggestions for Authors

The manuscript describes the production of beta glucan by Rhizopus oryzae M10A1 using liquid potato starch waste in submerged fermentation, and optimization of the culture media. The manuscript can be accepted for publication after minor revision. My comments are in below:

General comments:

  • The reference for '100 g' is unclear in several parts of the manuscript. Does it refer to biomass, the fermentation medium, or another material? For example, in line 403, the phrase "244.83 mg/100 g, respectively" suggests that something is being measured per 100 g, but the specific reference is not clear. Please specify it for clarity.

Abstract:

-Line 22: “ which was high purity”  should be “which was of high purity”.

-Line 24: “with 39.84 and 16.61%” should be “with percentage of 39.84 and…”.

Introduction:

-Line 33 to 38: The paragraph is hard to follow. The English can be improved for clarity and fluency.

Materials and Methods:

-Line 117: “stored at -4°C”. Is there any reason that the liquid should store in -4 not -20°C ?

--2.2. Inoculum preparation: A saline solution or ringer solution are often used to maintain the osmotic balance. Is there any specific reason for using sterile water?

-2.3. Culture condition: Baffled shake flasks are usually used for fermentation. Is there any specific reason for using volumetric flask?

-2.4.determination of chemicalcompositions

-Line 148: what concentration of H2SO4 was used?

2.5. Determination of Biomass: How the biomass was separated from the solid parts in the medium.

2.10. Structural characterization:

-Line 227:” at 0.5%” what is this number?

-Line 233: C18 columns is usually used for non-polar materials. Please add the reference paper that used this method for analysis.

  1. Results and discussions:

Figure 2: The Y-axis for the biomass does not need to have such a high range; a scale from 0 to 20 would be sufficient for both a and b.

Figure 4: Please specify what the color gradient represents.

3.4.3.Structural characterization:

-Line 579-581: This information is provided in Table 6, so it would be better to omit it here.

Table 6: Please specify what monosugar was the reference for this molar ratio?

-Line 600: arabinose was repeated.

-Line 622-627: “It has been reported that high molecular weight β-glucans develop high viscosities and are less soluble, and their effect at the physiological level is that of dietary fiber, which is not beneficial from the point of view of health, and they do not have the ability to form gels”. The sentence can be improved for clarity and to better convey the author's intended message.

Comments on the Quality of English Language

The quality of English in the manuscript can be improved for better clarity.

Author Response

1. Summary

Thank you for taking the time to make revisions to the submitted manuscript, your comments have been taken into account and some clarifications have been made. The document has been updated with the changes.

2. Questions for General Evaluation

Reviewer’s Evaluation

Response and Revisions

The reference for '100 g' is unclear in several parts of the manuscript. Does it refer to biomass, the fermentation medium, or another material? For example, in line 403, the phrase "244.83 mg/100 g, respectively" suggests that something is being measured per 100 g, but the specific reference is not clear. Please specify it for clarity

Yes/Can be improved

The references of 100 g were clarified, only in the culture medium the measurements are made based on 100 mL, in the other determinations they are based on 100 g

3. Point-by-point response to Comments and Suggestions for Authors

Comments 1: Abstract:  -Line 22: “ which was high purity”  should be “which was of high purity”.

-Line 24: “with 39.84 and 16.61%” should be “with percentage of 39.84 and…”.

Response 1: We agree with this comment. Therefore, we have modified the abstract for greater fluidity, background and methodologies were briefly mentioned, however the methodologies were not described due to their length, these are described in the materials and methods section. Between lines 10 and 23 the following was placed: β-glucans from filamentous fungi are important for human health. There is limited research on polysaccharides from filamentous fungi, and no reports have been published regarding the optimization of culture media for the production of β-glucans from Rhizopus oryzae using liquid waste from potato starch processing. In this regard, the fermentation conditions for the production of β-glucans from Rizhopus oryzae M10A1 were optimized using the One Variable at a Time (OVAT) and Response Surface Methodology (RSM). The β-glucans were chemically characterized by determination of moisture, nitrogen, protein, fat, ash and total carbohydrates. The color, molecular weight, β-glucan content, monosaccharide composition, structural and conformational characteristics were assessed by colorimetry, Gel Permeation Chromatography, High-Performance Liquid Chromatography and Fourier Transform Infrared Spectroscopy respectively. The microbial indicators, mesophilic aerobes, molds, yeasts and Escherichia coli were quantified following ISO standard protocols. Optimization indicated that supplementation with 0.8% (w/v) glucose and ammonium sulfate enhanced heteroglycan production (3254.56 mg/100 g of biomass). The β-glucans exhibited high purity, a light brown color, a molecular weight of 450 kDa, and a composition predominantly consisting of glucose and galactose. These findings suggest that β-glucans from Rhizopus oryzae M10A1 could be used for food and health applications.

Comments 2: Introduction:

-Line 33 to 38: The paragraph is hard to follow. The English can be improved for clarity and fluency.

Response 2: Agree. We have modified the introduction. The writing has been improved for better reading fluency.

Comments 3: Materials and Methods:

-Line 117: “stored at -4°C”. Is there any reason that the liquid should store in -4 not -20°C ?

Response 3: Agreed. The temperature used was -20 °C, this was corrected, in line 102 it says: Processing Factory, homogenized, and stored at -20 °C until it was required.

Comments 4:

-2.2. Inoculum preparation: A saline solution or ringer solution are often used to maintain the osmotic balance. Is there any specific reason for using sterile water?

Response 4: Bibliographic references indicate that distilled water and saline solution can be used to prepare the inoculum, since these are fungi, the osmotic balance is not affected in a short time, as it could be in bacteria.

Comments 5:

-2.3. Culture condition: Baffled shake flasks are usually used for fermentation. Is there any specific reason for using volumetric flask?

Response 5: In the fermentation, Erlenmeyer flasks and not volumetric flasks were used, it was corrected in line 113, which says the following: Volumetric flasks with a capacity of 250 mL… Baffled shake flasks were not used because they break the hyphae being cultivated in the liquid medium and we have observed that they form pellets instead of biomass.

Comments 6:

-2.4.determination of chemical compositions

-Line 148: what concentration of H2SO4 was used?

Response 6: The concentration of H2SO4 was placed in the manuscript. In the line 133 and 134 say: 15 mL of H2SO4 (concentrated sulfuric acid)

Comments 7:

2.5. Determination of Biomass: How the biomass was separated from the solid parts in the medium

Response 7: There were no solid elements in the culture medium from which the biomass had to be separated. Because the medium was liquid waste from starch processing, very few solids existed and part of them were consumed in the growth of the biomass. Therefore, it is indicated in section 2.6 that the biomass was washed with distilled water until the residues of the culture medium were removed.

Comments 8:

2.10. Structural characterization:

-Line 227:” at 0.5%” what is this number?

Response 8: In the manuscript it was placed (w/v) on line 216 it says: samples at a concentration of 0.5% (w/v)

Comments 9:

-Line 233: C18 columns is usually used for non-polar materials. Please add the reference paper that used this method for analysis.

Response 9: The reference for this paragraph is 31 as mentioned at the beginning of this section.

Comments 10:

Results and discussions:

Figure 2: The Y-axis for the biomass does not need to have such a high range; a scale from 0 to 20 would be sufficient for both a and b.

Response 9: Agreed. The following changes were made:

The axes of the Figure 2 were modified for better understanding.

Comments 11:

Figure 4: Please specify what the color gradient represents.

Response 10: Agreed. Information about the color gradient was incorporated. In Figure 4 the following was specified: As evidenced by the response surface (3D Graph) and contour plots (2D Graph), changes in color are indicative of varying combinations of ammonium sulfate and glucose concentrations and their impact on the production of β-glucans. Specifically, the intensity of the red color, observed at a concentration of 0.8%, indicates the highest polysaccharide concentration obtained.

Comments 12:

3.4.3.Structural characterization:

-Line 579-581: This information is provided in Table 6, so it would be better to omit it here.

Response 12:

The information in the writing corresponding to Table 6 was eliminated according to the suggestions.

Comments 13:

Table 6: Please specify what monosugar was the reference for this molar ratio?

Response 13:

Regarding the monosugar used, between lines 223 and 225 of the materials and methods section the standard sugars used are indicated.

Comments 14:

-Line 600: arabinose was repeated.

Response 14: Regarding the arabinose repeated in the writing, it was eliminated, In line 561 says: The major monosaccharide was glucose, followed by mannose, rhamnose, arabinose, galactose and fucose [55, 56].

Comments 15:

-Line 622-627: “It has been reported that high molecular weight β-glucans develop high viscosities and are less soluble, and their effect at the physiological level is that of dietary fiber, which is not beneficial from the point of view of health, and they do not have the ability to form gels”. The sentence can be improved for clarity and to better convey the author's intended message.

The paragraph can be improved for clarity and to better convey the author's intended message. The wording was modified, on lines 587 to 591 it says: From an industrial perspective, another important aspect is the relationship between molecular weight, viscosity, solubility and gel formation. It has been documented that high molecular weight β-glucans exhibit elevated viscosities and diminished solubility. Their physiological impact is analogous to that of dietary fiber, which is advantageous from a health perspective. Moreover, these compounds lack the capacity to form gels [61].

3. Response to Comments on the Quality of English Language

Point 1: The quality of English in the manuscript can be improved for better clarity.

Response 1: The English was revised and improved throughout the text. The entire manuscript was changed to allow for better understanding of the information.